# *Sideritis elica*, a New Species of Lamiaceae from Bulgaria, Revealed by Morphology and Molecular Phylogeny

**DOI:** 10.3390/plants11212900

**Published:** 2022-10-28

**Authors:** Ina Aneva, Petar Zhelev, Georgi Bonchev

**Affiliations:** 1Bulgarian Academy of Sciences, 1, 15 November Str., 1040 Sofia, Bulgaria; 2Department of Dendrology, University of Forestry, 10 Kliment Ohridski Blvd., 1797 Sofia, Bulgaria; 3Institute of Plant Physiology and Genetics, Bulgarian Academy of Sciences, Acad. Georgi Bonchev Str., Bl. 21, 1113 Sofia, Bulgaria

**Keywords:** medicinal plants, phenotypic variation, cryptic species, taxonomy

## Abstract

*Sideritis elica,* from the Rhodope Mountains, is described as a species new to science. Results of a detailed morphological analysis were combined with the data of molecular analyses using DNA barcoding as an efficient tool for the genetic, taxonomic identification of plants. The combination of morphological features distinguishes the new species well: Its first three uppermost leaf pairs are significantly shorter and wider, the branchiness of the stems is much more frequent, the whole plant is much more lanate, and it looks almost white, as opposed to the other closed species of section Empedoclia, which look grayish green. The molecular analysis, based on the rbcL and trnH-psbA regions, supports the morphological data about the divergence of *Sideritis scardica* and *Sideritis elica*. The studied populations of the two taxa were found to be genetically distant (up to 6.8% polymorphism for trnH-psbA) with distinct population-specific nucleotide patterns, while no polymorphism in the DNA barcodes was detected within the *Sideritis elica* population. The results confirm the existence of a new species called *Sideritis elica,* which occurs in the nature reserve Chervenata Stena, located in the northern part of the Central Rhodope Mountains. There were only 12 individuals found in the locality, which underlines the necessity of conservation measures.

## 1. Introduction

Genus *Sideritis* (Lamiaceae, Lamioideae) comprises more than 150 species distributed in the temperate and tropical areas of the Northern Hemisphere [1,2] and subdivided into two subgenera: *Sideritis* and *Marrubiastrum* (Moench.) Mendoza-Heuer. Southeastern Europe and the Eastern Mediterranean, with about 50 species, represent a center of diversity, particularly of section *Empedoclia* (Rafin.) Bentham of the subgenus *Sideritis*, with 45 species in Turkey [3], and about 10 species in Greece and the Balkans in general [4,5,6]. The number of species depends on their taxonomic treatment and concepts, which are not straightforward because of the high level of polymorphism, including ecotype diversity and hybridization among the species [7].

In Bulgaria, *Sideritis* is represented by four species, two of them belonging to section *Hesiodia* Bentham (*S. montana* L. and *S. lanata* L.), and two belonging to section *Empedoclia* (*S. scardica* Griseb. and *S. syriaca* L.) [8]. All but *S. montana* are considered rare and endangered species in need of measures for conservation [9,10,11]. Moreover, *S. scardica* and *S. syriaca* are considered essential medicinal plants and are subject to cultivation. While the two species of section *Hesiodia* are discrete and well-distinguishable, there are still some taxonomic uncertainties within the species of section *Empedoclia* [7].

A typical distribution pattern of *Sideritis* species of section *Empedoclia* is the high percentage of endemism [3,12,13]. *S. scardica* is endemic to the Balkan Peninsula, while *S. syriaca sensu lato* is believed to have wider distribution [4]. Both species naturally occur exclusively on limestone although they can be cultivated in a broader range of soil pHs [14].

A recent morphometric study on the Bulgarian populations of *Sideritis* spp. [15] revealed that the individuals in one population, treated as *S. scardica,* demonstrate substantial differences from those of the remaining populations. This population represents a distinct taxon, which can be very well-distinguished based on morphological traits.

Modern plant taxonomy could benefit substantially by the application of different DNA barcoding techniques [16,17,18,19,20], which reveal the phylogenetic relationships among the taxa and facilitate taxonomic decisions. This is especially important in cases of so-called “cryptic species”, i.e., species demonstrating low morphological, but considerable genetic, differences [21,22,23,24]. Genetic markers, including DNA barcoding, have been successfully applied to the study of different aspects of genetic diversity and the evolutionary relationships between different medicinal plants of the family Lamiaceae [25,26,27], including the species of genus *Sideritis* [28,29]. The application of DNA barcoding markers has been proven to be an important tool for various genetic and systematic studies.

The objective of the present study was to combine existing information from the morphological and morphometric studies with the utility of DNA barcoding on a local scale in discriminating *Sideritis* species. Four of the most frequently used cpDNA barcoding regions, matK, rbcL, trnH-psbA, and ITS1/ITS2, were used in our study. Both matK and rbcL have been selected as core barcodes by the Consortium for the Barcode of Life (CBOL) Plant Working Group (PWG), and ITS/ITS2 and trnH-psbA have been suggested as supplementary loci. In addition, the utility of these markers has been explored and discussed in previous studies on representatives of the family Lamiaceae [25,26,27,28,29]. Our last, but very important, objective was to provide a description of this new species of genus *Sideritis*, sect. *Empedoclia*.

## 2. Results

### 2.1. Morphological and Morphometric Data

The full results of the statistical analysis are presented in [15], and they revealed that the population Chervenata Stena (CHE), classified as *S. scardica,* differed significantly from the remaining ones of the same species. The distinct features of CHE are the higher-flowering stems, densely lanate stems and leaves, and different shape of the uppermost 1–3 pairs of leaves. In more detail, the height of the flowering stems of the individuals here reached 40–60 cm, as opposed to 20–40 cm in other populations. Leaf indumentum was clearly different, too—it was densely white and lanate in the individuals in the CHE population, as opposed to the grayish green indumentum of the leaves of plants from the other populations (Figure 1 and Figure 2). A very indicative trait was the ratio length/width of the 2–3 uppermost leaf pairs. In the individuals from the CHE population, this ratio ranged between 2 and 3, while in the other populations, it ranged between 3 and 6 (Figure 3). These results are summarized in Table 1.

The applied cluster and principal component analyses clearly separated the population Chervenata Stena from the remaining populations of both *S. scardica* and *S. syriaca* (see [15]).

The juvenile seedlings of *S. scardica* from Trigrad and *S. elica* differed clearly in their pubescence although both localities are situated in the same mountain range (Figure 4). It can be seen in the figure that the cotyledons and the stems of *S. elica* plants are densely covered by trichomes, while in *S. scardica* there are only single trichomes, located mostly on the stem.

### 2.2. DNA Barcoding

Five specimens from Chervenata Stena and six from Slavyanka Mountain were analyzed using DNA barcode markers. The efficiency of PCR amplification and sequencing for the *Sideritis* specimens was 100%, despite the fact that the sequence quality of the amplified products was low for *matK* and ITS primers. On the other hand, DNA barcodes for regions *rbcL* and *trnH-psbA* clearly indicated that *Sideritis* specimens from the two floristic regions are separated genetically, with a lack of polymorphism for samples from the floristic region Chervenata Stena (Figure 5). The divergence was visualized by the presence of insertions and deletions (Indels) and/or as single nucleotide polymorphisms. The trnH-pasbA region was the most informative, with Indels and SNPs found to be population-specific. The rbcL marker showed a unique SNP (CA/CG), which is also population-specific (Figure 5, black box).

The genetic divergence, both within and between populations (Table 2), showed that, when taking into account the total polymorphic sites, including insertions/deletions, the value of genetic divergence accounts for 6.8% for trnH-psbA marker (35/508, data not shown).

In order to address the taxonomic status of *Sideritis* specimens from both floristic regions, we performed a BLAST analysis of the region-specific consensus sequence (e.g., S10 and S18) against the NCBI database accessions (for DNA barcode trnH-psbA) and NCBI database + BOLD database (for DNA barcode rbcL). The taxonomic assignment (Figure 6) shows that the samples from Slavyanka Mountain belongs to *Sideritis scardica,* forming a cluster with other accessions mainly of this species from the NCBI database. The sample representative for the Chervenata Stena population, although close to *Sideritis scardica* (Figure 5, gene rbcL), is slightly genetically distant and clustered separately. As revealed by the DNA barcode trnH-psbA, the specimens from the Rhodope Mountains clustered predominantly with *Sideritis sipylea* Boiss.

### 2.3. Taxonomic Implications

The combined results of the morphometric and DNA barcode studies showed that the population of Chervenata Stena differs from the other populations of *S. scardica* and suggest that it represents a distinct species.

## 3. *Sideritis elica* Aneva, Zhelev and Bonchev sp. *nova*

Type: BULGARIA: Rhodope Mountains: Chervenata Stena Biosphere Reserve, 41°54.163′ N 24°53.699′ E, 1350–1470 m ASL, on dry calcareous rocky slopes, that are part of Natura 2000 habitat 5210, arborescent matorral, with *Juniperus* spp., and the locality is near to 9530 *(Sub)-Mediterranean pine forests with endemic black pines habitat, 4 July 2012, Ina Aneva and Petar Zhelev (holotype SOM 172837; isotype SOM 172840).

*Etymology:* The new species is named *Sideritis elica*, in honor of Elka Aneva (the mother of Ina Aneva) in recognition of her botanical enthusiasm and inspirational help during all of the field studies.

*Diagnosis*: *Sideritis elica* differs from *S. scardica* according to the following features: the whole plant is densely whitish lanate (vs. densely pubescent), the upper three leaf pairs are 20–40 mm long and 10–15 mm wide (vs. 30–40–70 mm long and 6–12 mm wide); the ratio of length/width is 2 to 3 (vs. 3 to 6).

*Description*: Perennial herb, with well-developed rhizome, sometimes woody at the base. Stems 40–60 (70) cm long, simple or branched, densely white lanate, erect or ascending. Basal leaves 30–70 mm long and 10–20 mm wide, narrowed to the base. Stem leaves opposite, slightly serrate, densely white lanate, sessile, leaves of the upper 2–3 pairs ovate-lanceolate, 20–40/10–15 mm. All leaves obtuse at the apex, ratio length/width of the upper 1–3 leaf pairs 2 to 3. Inflorescence simple, rarely a few branched. Verticillasters 6–8, bracts 14–16/10–12 mm (Figure 1d,e; Figure 2c,d).

There is some overlap in the morphological and quantitative characteristics, allowing for the hypothesis that the new species possesses some characteristics of a cryptic species. Along with the morphological differentiation, there are clear differences in the DNA barcoding markers *trnH-psbA* and *rbcL*, as shown by the phylogenetic analysis (Figure 6).

*Sideritis elica* is a narrow endemic, occurring on limestone in the northern part of the Central Rhodope Mountains in Bulgaria. The species grows in the periphery of a mixed forest *Pinus nigra* Arn. and *Ostrya carpinifolia* Scop. Some individuals also occur within *Juniperus deltoides* R.P. Adams (*J. oxycedrus* L.) scrub, at an altitude of 1180–1210 m.

The population size is low; the exact number of individuals is not known because no detailed inventory has been performed so far, but a rough estimate allows for the conclusion that there are not more than ~20 individuals in the locality (only 12 individuals were registered in area of 0.5 ha). The individuals have relatively large sizes (in contrast to those of S. scardica in their natural localities): one individual with an average size of 120 × 80 cm, three individuals with an average size of 90 × 60 cm, two individuals of 80 × 50 cm, two of 60 × 40 cm, and four of 40 × 30 cm. The total area occupied by the individuals is 4.34 m^2^. Applied to the entire area of the natural locality (5000 m^2^), this equates to a projective coverage of 0.087%. The population is in a critical state; both the area and the projective cover of the species have very low values. The anthropogenic factor has a great influence due to the proximity of the locality to the Martsiganitsa Hut. Currently, no specific threats have been identified; however, the low population size provides a risk per se, related either to direct damage, or to genetic erosion of the population.

## 4. Materials and Methods

Morphologic and morphometric studies were performed on eight populations, representing six populations of *S. scardica* and two populations of *S. syriaca* [15]. Particular attention was given to the population Chervenata Stena (CHE) from the Rhodope Mountains (41°53′ N 24°51′ E, 1300 m ASL). Details of the studies are presented elsewhere [15]. In addition, we collected seeds of *S. scardica* from Trigrad (Rhodope Mountains) and of *S. elica* from the Chervenata Stena locality in order to compare the morphology of juvenile seedlings. The seeds were sown under the same controlled conditions.

A dataset of 11 specimens of *Sideritis* from the locality Chervenata Stena (No. 9–13) and the Slavyanka Mountain (No. 15–20) were used for DNA barcoding analysis.

### DNA Extraction, PCR Amplification, and Sequencing

Genomic DNA was extracted by using an Invisorb^®^ Spin Plant Mini Kit (Invitek Molecular GmbH, Berlin, Germany) following the instructions of the manufacturer. DNA quality and quantity were measured by a NanoDropTM Lite Spectrophotometer (Thermo Fisher Scientific). The genetic diversity of the samples was evaluated based on sequences of universal barcodes for plants: nuclear ribosomal internal transcribed spacer (ITS), ribulose-1,5-bisphosphate carboxylase/oxygenase large subunit (rbcL) gene, maturase K (matK) gene, and psbA-trnH intergenic spacer. The sequences of the used primers (synthesized by Microsynth) and the PCR conditions are presented in Table 3. PCR amplification was performed in 20-µL reaction mixtures containing approximately 30 ng of genomic DNA, 1 X PCR buffer, MgCl_2_ (2.0 mM for ITS and matK, and 2.5 mM for rbcL and trnH-psbA), 0.2 mM of each dNTP, 0.2 µM of each primer, and 1.0 U Taq DNA Polymerase (Solys Biodine). The quality of the PCR products was checked on 1% agarose gel containing GoodViewTM staining dye. Successful amplicon products were sequenced in both directions by Microsynth (Germany), using the same primers used for the PCR amplification.

Candidate DNA barcode sequences for each barcode region were aligned via MEGA-X, and consensus sequences were subjected to further analyses using the software package Geneious. The phylogenetic trees were constructed using the Jukes–Cantor genetic distance model [30], and the UPGMA tree-building method. Evolutionary divergence was tested using the Tamura 3-parameter model [31] implemented in MEGA-X software [32].

Taxonomic assignment of the *Sideritis* specimens was performed through BLAST analyses in Geneious against publicly available accessions in NCBI. The estimates of within population and between population divergence were calculated in MEGA-X [32].

## 5. Conclusions

In this study, we performed a morphological and DNA barcoding analysis of representatives of *Sideritis scardica* populations from two geographically distant floristic regions in Bulgaria. The *Sideritis* population from the reserve Chervenata Stena (CHE) was found to be phenotypically distinct from *Sideritis scardica*. This allowed us to state that the population from the reserve represents a new a new species we called *Sideritis elica* Aneva, Zhelev and Bonchev. The genetic divergence between *S. scardica* and *S. elica* was supported based on rbcL and trnH-psbA markers. The data has two main implications. First, our study implies that eco-geographical and demographic conditions enhance genetic diversification and occasionally the speciation within the genus *Sideritis*. Second, our study highlights the importance of the DNA barcoding method to unravel patterns of genetic variability at species in support of classical morphological approaches. Further studies on a larger set of *Sideritis* populations could give deeper insight into the ecological dynamics of this endemic genus of high medicinal value for Bulgaria, with practical implications for its conservation.

## Figures and Tables

**Figure 1 plants-11-02900-f001:**
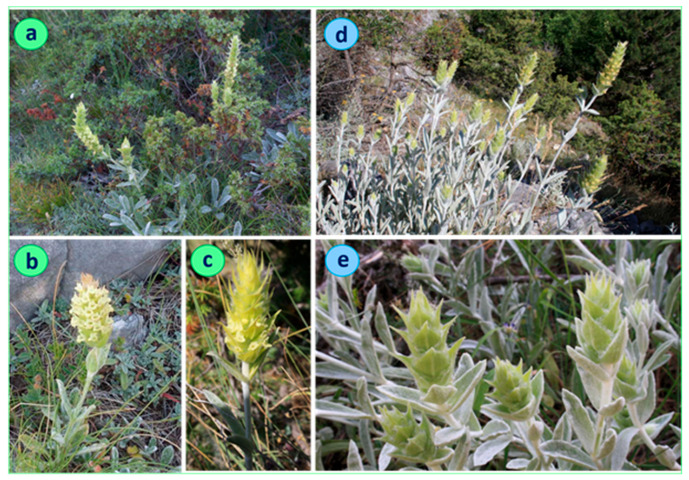
*Sideritis scardica*: live individuals from the Slavyanka Mountain (**a**–**c**) and *Sideritis elica* from Chervenata Stena (**d**,**e**)—upper part of the plant.

**Figure 2 plants-11-02900-f002:**
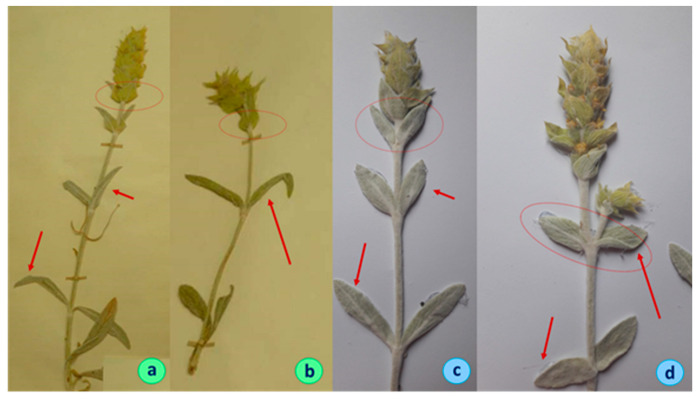
Herbarium specimens: *S. scardica* (**a**,**b**) and *S. elica* (**c**,**d**).

**Figure 3 plants-11-02900-f003:**
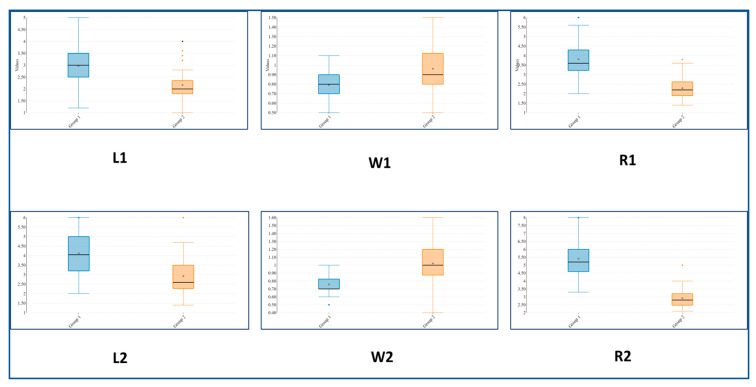
Boxplot for the most discriminant morphological characters. L1, W1, R1: length, width, and ratio of length/width of the uppermost leaf pair; L2, W2, R2: the same traits for the second leaf pair.

**Figure 4 plants-11-02900-f004:**
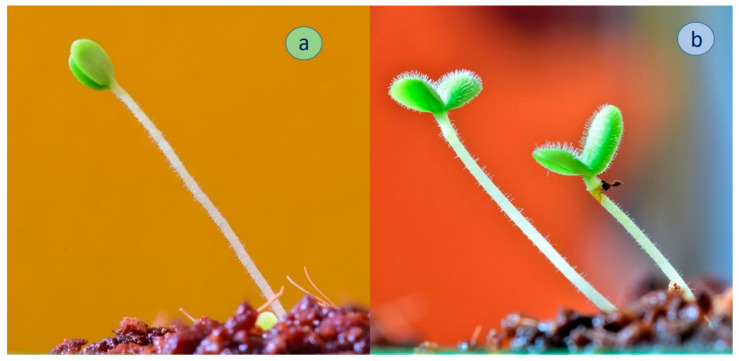
Juvenile plants of: (**a**) *Sideritis scardica* and (**b**) *Sideritis elica.*

**Figure 5 plants-11-02900-f005:**
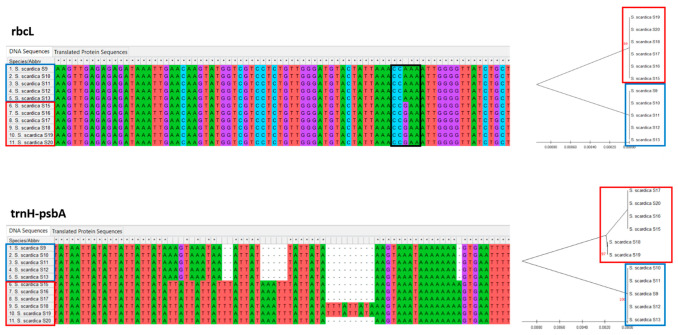
Genetic discrimination of *Sideritis* specimens from two distant floristic regions based on DNA barcodes trnH-psbA and rbcL, visualized both as a sequence alignment (partial sequences) and as a phylogenetic tree. The only single nucleotide polymorphism (SNP) between specimen populations based on the rbcL marker is indicated by a black box. The specimens from Chervenata Stena and Slavyanka Mountain are marked with blue and red boxes, respectively.

**Figure 6 plants-11-02900-f006:**
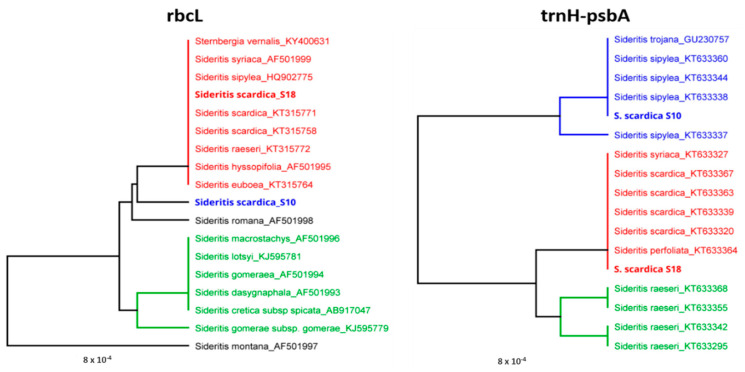
Taxonomic assignment of region-specific *Sideritis scardica* specimens (S10 and S18, in bold) against BOLD database accessions (trnH-psbA) and BOLD+NCBI (rbcL). The trees were constructed in Geneious software using the genetic distance model of Jukes–Cantor and the tree construction method of UPGMA.

**Table 1 plants-11-02900-t001:** Comparison of the key morphological characters of *S. scardica* and the new species, *S. elica*.

Characters	*Sideritis scardica*	*Sideritis elica*
Stem height	20–30 (40) cm	40–50 (60) cm
Stem pubescence	Densely pubescent	Densely white lanate
Length of the upper leaves (cm)	3–3.5	2–2.5
Width of the upper leaves (cm)		
Ratio L:W of the upper 1–3 pairs of leaves	3–6	2–3
Leaf indumentum	Pubescent; gray and white trichomes	Densely lanate; white trichomes
Leaf color (mature leaves)	Grayish green	White; lanate

**Table 2 plants-11-02900-t002:** Estimates of evolutionary divergence (D) between populations (BP) and within populations (WP) Chervenata Stena and Slavyanka Mountain. The number of base substitutions per site, obtained by averaging the overall sequence pairs between groups, are shown, along with standard errors. Analyses were conducted using the Tamura 3-parameter model implemented in MEGA-X.

	Barcoding Markers			
	rbcL		trnH-psbA	
	WP			
	D	SE	D	SE
Chervenata Stena	0.00	0.00	0.00	0.00
Slavyanka Mountain	0.00	0.00	0.001	0.001
	BP			
	0.002	0.002	0.018	0.006

**Table 3 plants-11-02900-t003:** Oligonucleotide primers used for amplification of DNA barcode regions, and the respective PCR conditions.

Barcode Region	Primers	Primer Sequences 5′-3′	PCR Conditions
matK	matK-1RKIM-f	ACCCAGTCCATCTGGAAATCTTGGTTC	95 °C 5 min (95 °C 30 s, 51 °C 50 s, 72 °C 1 min)—35 cycles 72 °C 7 min
	matK-3FKIM-r	CGTACAGTACTTTTGTGTTTACGAG	
rbcL	rbcLa-F	ATGTCACCACAAACAGAGACTAAAGC	94 °C 4 min (94 °C 30 s, 55 °C 30 s, 72 °C 1 min)—35 cycles 72 °C 10 min
	rbcLajf634R	GAAACGGTCTCTCCAACGCAT	
trnH-psbA	trnH-F	CGCGCATGGTGGATTCACAATCC	94 °C 4 min (94 °C 30 s, 55 °C 30 s, 72 °C 1 min)—35 cycles 72 °C 7 min
	psbA3_r	GTTATGCATGAACGTAATGCTC	
ITS	ITS F1	CCTTATCATTTAGAGGAAGGAG	94 °C 5 min (94 °C 30 s, 50 °C 30 s, 72 °C 1 min)—35 cycles 72 °C 5 min
	ITS 4	TCCTCCGCTTATTGATATGC	

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
