# Peer review of "Sideritis elica, a New Species of Lamiaceae from Bulgaria, Revealed by Morphology and Molecular Phylogeny"

_plants, 2022, doi:10.3390/plants11212900_

Round 1
Reviewer 1 Report
The work is well presented but there are some shortcomings which must be revised.
In the abstract methods must be briefly described.
Prominent morphological features and resemblance percentage by molecular analysis must be added in the abstract.
Add future perspective in the abstract.
Key words, capitalize first word of key words.
Add more details of the significance of phylogenetic techniques and DNA barcoding in second last paragraph of the introduction. The following studies could be cited. https://doi.org/10.1111/jse.12642, https://doi.org/10.3390/agronomy12051078,
Provide reasons why these ((matK, rbcL, trnH-psbA) and the ITS region) were used in this study.
What is its relation with this genus.
Line 59 previous studies provide references.
Line 65 provide and describe results of this study.
Figure 1 must be more clarify. Resolution is very poor.
Section DNA extraction should be cited with https://doi.org/10.2298/GENSR2002435A,
Conclusion must be extended also add future perspective.
Author Response
Dear Reviewer,
Thank you very much for your professional remarks. We fixed them all and now the manuscript looks really improved. We clarified the whole text and add the suggested references.
Reviewer 2 Report
The manuscript and images made it clear to me that this was an undescribed species. The scientific description and the explanation in the text made it possible and easy to visualize the plants and their habitat and to agree with the conclustions of the authors.
Author Response
Dear Reviewer,
Thank you very much for your professional look at our manuscript!
We fixed all mistakes and now the text is significantly improved.